# ITV-Pal programme: protocol of evaluation of the implementation of tech-volunteer programme in palliative care services

Pilar Barnestein-Fonseca,[1,2] Eva Víbora-Martín [ID] ,[1,3,4]
Inmaculada Ruiz-Torreras,[1,3,4] Helena Chapinal-Bascón,[1]
Maria Luisa Martín-Roselló,[1,4] Rafael Gómez-García [ID] [1,4]

## ABSTRACT

**Introduction** Volunteer support for patients and families at the end of life provides many benefits for the beneficiaries. New technologies could be a necessary resource in the accompaniment although, if there is little literature on palliative care volunteering in general, specifically on volunteering and new technologies, we find little information on the subject.
Therefore, the aim of this study is to implement and evaluate a training program for palliative care volunteers using new technologies in order to begin accompanying patients and families in hospital or at home.

**Methods and analysis** A mixed-method study design will be conducted. We will recruit 20 volunteers and 70 patients in two years. Intervention: training of volunteers in new technologies and volunteer accompaniment of patients/relatives using technologies. The control group will accompany patients as usual.

**Ethics and dissemination** Ethics approval for the ITV-Pal Programme project was granted by the Malaga Regional Research Ethics Committee. As new knowledge is gained from this project, findings will be disseminated through publications, presentations and feedback to clinicians who are participating in this study.

**Trial registration number** NCT04900103.

**Correspondence to**
Dr Eva Víbora-Martín;
evavibora@cudeca.org

## STRENGTHS AND LIMITATIONS OF THIS STUDY

⇒ The use of a mixed model that combines qualitative and quantitative methodology is the best way to evaluate the implementation of a new service, which is why it will be the one chosen in this study.

⇒ One of the limitations of the study is the effect of the intervention on recruitment and that this may result in different recruitment rates.

⇒ The main risk of this study is that we will not be able to attend to the different needs of the patients/families, although we will try to offer volunteers who can provide a better response to these needs.

## INTRODUCTION

How we care for the dying is perhaps the most pressing personal, social and public health issue of the 21st century. The experience of dying as an integral and acknowledged part of individual life, supported by the very best personalised care, is currently far from being standard practice in Europe and beyond.[1] Furthermore, it is estimated that by 2050, one in five people in the world will be over the age of 60[1] with an estimated 52 million people expected to die worldwide in 2030 from chronic diseases such as cancer, cardiovascular and renal diseases and others.[2] This poses a new challenge to the functional capacity and organisational structures of current healthcare systems.

Distress at the end of life is significantly associated with both physical/somatic and non-physical symptoms. For this reason, patient care cannot focus only on treating physical symptoms through medication. When a terminal illness disrupts a life, it leads to the loss of important relationships, as well as changing the structure of the patient's social network and the levels of relationship and support.[2] Dying is, therefore, also a social process.[3] The psychological distress due to feelings of anxiety, depression, loneliness and being a burden requires additional approaches to care.[4 5] Approaches that aim to provide supplementary, informal, lay support for patients dying at home, and their relatives, may introduce a valuable sense of 'community' and lead to important improvements in the care of the dying. Volunteer support represents an important and necessary community resource for patients and relatives to enhance the experience of living at the end of life and dying.[4 5] Although few studies exist, those few agree on the beneficial aspects of volunteering on patient and family satisfaction at the end of life.[4 5]

It is of paramount importance to develop innovative, economically viable interventions to support the care that dying patients and their relatives receive. In this way, the current health emergency situation and the confinement has revealed the isolation and loneliness that certain groups have already experienced for a long time, such as the elderly, immigrants or patients and caregivers.[6] New technologies (NT) may become a useful resource in this kind of situation and have already proven a benefit in the context of palliative care (PC). Healthcare professionals (HCPs) use information and communication technologies to facilitate health services for patients who have problems with access to the services, any disability which restricts movement or who live in a rural area.[7 8] In some studies, they found that the use of NT in consultation was as effective as face to face.[9–13] It was even more positive for caregivers, reducing their anxiety and increasing their quality of life.[14] In this context, the implementation of services that combine volunteer support and NT can be an innovative, economically viable intervention to support the care of dying patients and their relatives. In addition, these approaches bring volunteering closer to people who would like to offer their help but have little time or cannot be in the same place as the patient/relatives. In these cases, virtual volunteering could be a way to reach a larger population.[15]

Studies related to volunteering with NT mainly address the challenges of bringing these types of devices closer to the older population[16 17] or how volunteers perceive them as another resource to improve their relationship with the caregivers of patients,[18] but even so organisations are far from including such resources in their practice.

Technologies resources facilitate volunteer interaction and communication with the community, which is increasingly dependent on NT. Furthermore, volunteers with NT training will be able to help patients with daily practical tasks. NTs have also proven useful for therapeutic interventions. The patient can use them to review their biography, maintain social relationships with people who live far away or help them to build a legacy for their loved ones.[19 20]

Although volunteers contribute many hours to accompanying and supporting patients and their relatives, the review of bibliography found no sufficiently robust studies to merit inclusion, and even less when associating PC volunteering and NT. More research is needed to assess the impact of training and support of volunteers in PC. This study aims to analyse the technophobia (fear or aversion towards NT or complex devices) and technophilia (opposed to technophobia and is responsible for establishing the behaviour of adherence, often uncritically, to technological innovations) of patients, relatives, volunteers and health professionals. Once their knowledge of new NT has been assessed, a training programme for volunteers will be evaluated and a programme of accompaniment through technological devices will be implemented.

The aim of this study will be to evaluate the implementation of a volunteer training programme in the use of NT (specifically through smartphones and tablets) to support patients facing a life-threatening illness and their relatives. The specific objectives will be: (1) to explore the need and usefulness of NT from the point of view of patients, relatives, volunteers and HCP in PC and describe their technological profile; (2) to design a tech-volunteer curriculum and implement a tech-volunteer training programme and (3) to implement the tech-volunteer programme within a PC Home care service and Inpatient unit and assess its impact on the care provided.

## METHODOLOGY

The Research Ethical Committee of Málaga (25 February 2021) approved this study.

### Study design

The best approach to evaluate the implementation process is through mixed-method and the combination of quantitative and qualitative methods.[21] Basic quantitative measures of implementation may be combined with in-depth qualitative data to provide detailed understanding of intervention functioning on a small scale. Quantitative methods will measure key process variables and allow testing of prehypothesised mechanisms of impact and contextual moderators. Qualitative methods will capture emerging changes in implementation, experiences of the intervention and unanticipated or complex causal pathways, as well as generating a new theory.

#### A: Quantitative approach

► Pragmatic cluster randomised clinical trial to test efficacy (the unit of randomisation is the volunteer and the unit of analysis is the patient/relatives). A pragmatic cluster randomised controlled trial has shown to be the most appropriate design when the programmes are more focused on the organisational level.[22 23] For our study, the cluster design is based on two levels: the higher or second level is represented by the volunteer (over volunteer the intervention is conducted, ITVPal Programme: Tech-Volunteer Programme in Palliative Care Services), and the lower or first level being represented by the patients/relatives, who have agreed to participate and are going to receive the tech-volunteer care from volunteers of ITVPal Programme.

► Before-and-after design for satisfaction of volunteers and HPC with the intervention and its implementation to test effectiveness. The design involves evaluating the effects of a deliberate intervention (introduction of ITVPal Programme) by comparing the outcomes of study participants investigated before the intervention with those measured afterwards.

► Cost–utility study from the perspective of the funder with a time horizon of 2 years. A detailed cost analysis including costs for the adaptation of a volunteer

standard course to an ITVPal Programme as well as the recruitment and training costs of the volunteers will be performed.

## B: Qualitative approach

Interviews (individuals and groups) with HCP, volunteers and with key informants of patients/relatives to test at the beginning the need and usefulness of NT, and also during the implementation process to test changes and experiences. A narrative approach will be adopted, facilitating the production of 'rich' data regarding how participants make 'sense' of and engage with the ITVPal programme. These interviews will explore the thoughts and feelings engendered by access to this service, initial motivations, as well as thoughts and feelings about care delivery as a whole. Narrative interviewing enables the participants to tell their 'stories', prompted by an open question.

## Settings

The study will be conducted in a palliative care unit from both a home care and an inpatient unit of the CUDECA foundation.

The hospitalisation unit is a small hospice consisting of 10 rooms where patients at the end of life are cared for in cases of uncontrolled symptoms, for family respite or in their final days, among other situations. In this unit, volunteers come every afternoon to the rooms where they wish to offer accompaniment to patients and their relatives.

In addition, CUDECA foundation provides home care to more than 1500 patients a year through 7 care teams made up of a doctor, nurse, psychologist and social worker. In these cases, the volunteer accompanies patients and relatives who wish them to do so.

## Participants

A total of 140 patients and/or relatives will be selected from the home care and inpatient unit of CUDECA foundation where the ITV-Pal programme will be implemented. This sample size is enough to detect a mean difference of 3 in the HAD scores between groups of 25%, with a power of 80% and a confidence level of 95%.[24] Sample size was adjusted according to the standard criteria for cluster randomised trials, using the design effect (DE) of 1.3. The DE was calculated as follows: $DE=1+(nc-1)\times ICC$ (where nc is the mean number of individuals in the cluster and ICC the intracluster correlation coefficient). The ICC in the present work was considered to be 0.05 and the mean cluster size was assumed to be 7. A potential loss of 20% was estimated. Therefore, 140 patients/relatives and 20 volunteers are required.

The inclusion criteria will be: patients and relatives cared for by a specialised PC team; to be over 18 years of age; consenting to participate in the study by signing the informed consent form. Subjects who have specific conditions reducing their physical ability to use the devices, for example, visual, hearing or motor impairments will be excluded from the study.

The HCP involved in these services will be included to measure the success of implementing the programme.

The inclusion criteria for volunteers will be an interview with the Volunteer Department and having completed basic care training to be able to carry out their work. Also, a commitment to participate in the study for 1 year. For this reason, more volunteers will be trained as needed during the 2 years of implementation.

## Randomisation

The radomisation will take place in the higher or second level represented by the volunteer (over volunteer the intervention is conducted: ITVPal Programme) using a block randomisation technique. Each block is formed by four volunteers among whom the two study arms will be uniformly distributed. Once the blocks have been created, they will be distributed using a sequence of random numbers and the final list of volunteers allocations will be created. The final list of volunteers allocations will be guarded by the principal investigator of the project, and he will inform the allocation of each volunteer included in the trial.

## Patient and public involvement

The design of this protocol did not involve patients.

## Intervention

A complex intervention will be carried out in two phases.
1. First phase: training of volunteers in NT. Volunteers will be trained in the use of NT and their interconnectivity as tools to support their work as volunteers.
   The topics that will be covered in the volunteer training are the role of the volunteer, use of NT, communication strategies, complicated situations, self-care, summary and closure of the training. All of this is divided into 10 sessions, each lasting approximately 3 hours. This training will be integrated into the specific training programme for PC care volunteers.
   The control group, on the other hand, will receive the standard training that the care volunteers have received so far.
2. Second phase: volunteer accompaniment of patients/relatives. During the activities carried out by the volunteer with the patient for accompaniment and leisure. It will have the support of a technological device to be used according to the needs of the patient/relatives (eg, internet searches on a topic of interest, video calls, phone calls, virtual visits) It will be an outcome of the study to know what the device has been used for.
   The control group will accompany patients without using the NT as usual.

## Variables

Sociodemographic variables such as date of birth, gender, educational level, nationality, spiritual beliefs and employment status will be assessed in all groups.

Specifically, relatives will be asked about their relationship with the patient. Volunteers will be questioned about their time of collaboration and whether they have had

personal experience with palliative patients and finally, health professionals will be asked specifically about their work experience in PC.

In the following section, the different tools of assessment that each group will have to complete will be indicated:

### Patients

▶ The ICECAP Supportive Care Measure (ICECAP-SCM) [25] has been developed as a tool for use in economic evaluation conducted in an end of life setting for patients. The questionnaire covers the attributes of: choice, love and affection, physical suffering, emotional suffering, dignity, being supported and preparation with the lowest score being 1 and the highest score being 4.

▶ EuroQol-5D-5L, [26] a generic, brief and easy-to-use questionnaire designed to describe and assess the current health-related quality of life of the person completing the questionnaire. Primary outcome: The descriptive system comprises five dimensions: mobility, self-care, usual activities, pain/discomfort and anxiety/depression. Each dimension has five levels: no problems, slight problems, moderate problems, severe problems and extreme problems. The patient is asked to indicate his/her health state by ticking the box next to the most appropriate statement in each of the five dimensions.

▶ Distress thermometer[27] was developed as a simple tool to effectively screen for symptoms of distress. The instrument is a self-reported tool using a 0–10 rating scale.

▶ Edmonton Symptom Assessment Questionnaire[28] used to rate the intensity of nine common symptoms experienced by cancer patients, including pain, tiredness, nausea, depression, anxiety, drowsiness, appetite, well-being and shortness of breath where 0 is that the symptomatology does not affect him/her at the moment and 10 with the highest intensity.

▶ Tech-PH[29] an instrument for measuring older people's attitudes toward technology: It was created from the six items in the two factors, techEnthusiasm and techAnxiety. TechPH could be interpreted on a five-point response scale, ranging from 1 (fully disagree) to 5 (fully agree), where the higher the index indicates a higher level of technophilia.

▶ The Questionnaire on the Frequency of and Satisfaction with Social Support (QFSSS)[30] was designed to assess the frequency and degree of satisfaction with perceived social support from different sources in relation to three types of support: emotional, informational and instrumental. Both frequency and satisfaction are measured on a Likert scale of 1–5.

▶ Hospital Anxiety and Depression Scale (HADS)[31] instrument to measure anxiety and depression in a general medical population of patients. The lowest score being 0 and the highest score being 3.

### Relatives

▶ ICECAP Capability Measures (ICECAP-CPM)[32] has been developed as a tool for use in economic evaluation conducted in an end of life setting for relatives. The questionnaire covers the attributes of: good communication, privacy and space, emotional support, practical support, being able to prepare and cope and being free from emotional distress related to the condition of the decent with the lowest score being 1 and the highest score being 5.

▶ EuroQoL-5D-5L.[26]

▶ Distress thermometer.[27]

▶ TechPH.[29]

▶ QFSSS.[30]

▶ HADS.[31]

▶ Care of the Dying Evaluation (iCODE)[33] is a self-completion post-bereavement questionnaire, based on the key components of best practice in end of life care. It assesses quality of care for the dying through 32 main questions, reflecting core PC principles.

### Volunteers

▶ ICECAP-Adults (ICECAP-A)[34] is a measure of capability for the general adult (18+) population for use in economic evaluation. The questionnaire covers the attributes of: attachment (an ability to have love, friendship and support), stability (an ability to feel settled and secure), achievement (an ability to achieve and progress in life), enjoyment (an ability to experience enjoyment and pleasure) and autonomy (an ability to be independent) with the lowest score being one and the highest score being 4.

▶ EuroQoL-5D-5L.[26]

▶ Distress thermometer.[27]

▶ Satisfaction with volunteering; Motivation (VFI)[35] a 30 item measure of motivations to volunteer. Respondents answer each item on a 7-point scale ranging from 1 (not at all important/accurate) to 7 (extremely important/accurate).

▶ TechPH.[29]

### Healthcare professionals

▶ TechPH.[29]

Focus groups or in-depth interviews will be conducted with all groups at different stages of the study to assess the implementation of the study.

Process evaluation: Service Improvement Metrics: Process evaluation measures will be collected to assess the 'success' of the implementation process and how the service is being delivered. Data will be collected to assess the following: fidelity, dose and reach. Fidelity: Success of the Infrastructure. Dose: identification of patients, eligibility and take up of the service and whether the intervention was delivered as intended. Reach: (spread of implementation) quantifying the intervention implemented with the number of patients eligible for the service, number of patients who accept the service,

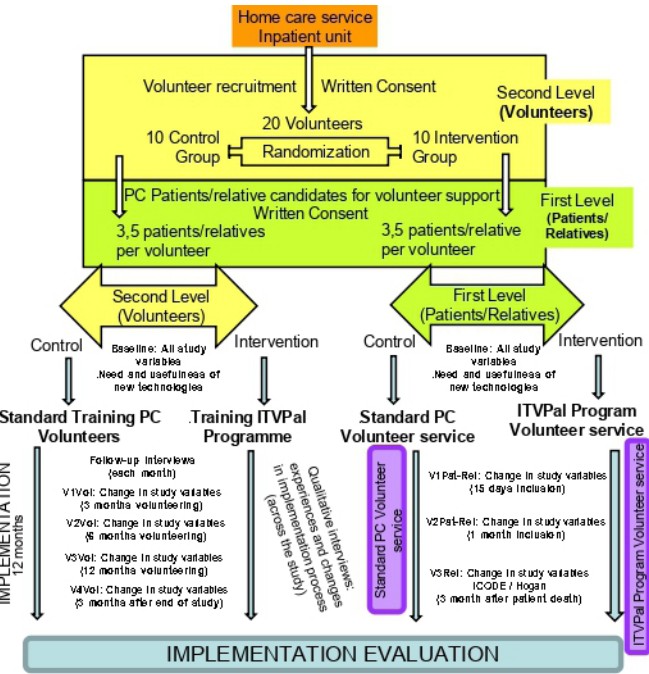

**Figure 1** Schematic of patient enrolment and data collection in the ITV-PAL study. Own elaboration. PC, palliative care.

number of wards with 'eligible' patients and number of wards where the service is implemented.

## Data collection and follow-up

The recruitment of the volunteers began in March 2021. When they finished their training, the new service started in September 2021 which was when patients/relatives recruitment started. Volunteer recruitment will be active during the whole study every 3 months. Recruitment is expected to end in September 2023.

The care teams offer the accompaniment of a volunteer to the patient/relatives. If they accept the accompaniment and participate in the study (if they want the accompaniment of a volunteer but do not want to participate in the study, the volunteer will be offered anyway). Once the informed consent to participate in the study has been signed, the data collection will proceed as described below. Further details can be seen in figure 1.

Patients/relatives who consent to participate (intervention and control group) will be invited to fill in at least three short questionnaires: at the time of recruitment for this study, 15 days after recruitment following the volunteer service experience (if hospitalised), and every month as long as the patient remains under PC care or until death. During the follow-up, patients and relatives will be asked if there is any change in their condition with respect to the questions asked at baseline. In this way, costly follow-up will be avoided. A maximum of 5 patients/relatives will also be invited to participate in an in-depth interview at various points during the implementation of the programme.

Volunteers will spend 1 year accompanying patients in ITV-Pal. In the following year, there will be other newly trained volunteers joining the project. During their time,

they will fill in five questionnaires: baseline, 3, 6 and 12 months following volunteering, and 3 months after. A maximum of five volunteers and five HCP with direct experience of the service will be approached for inclusion in in-depth interviews to explore their experiences during the implementation process and at the end of this study.

## Data analysis
### Quantitative data

Intention-to-treat principle: Descriptive statistics will be used to summarise subject characteristics (age, gender, education, religion, socioeconomic status, marital status, place of residence) and the process evaluation. Basal comparison between groups for independent samples: Student's t-test or $\chi^2$ test; for related samples: analysis of variance for repeated measures to analyse changes within groups. Multilevel analysis will be used for main and secondary outcomes. A multilevel modelling approach will be used to examine the difference in the primary endpoint between patients/relatives, taking account of clustering effects at volunteers level. All statistical tests will be two sided and considered significant if $p < 0.05$. Repeated measurements of analyses of variance will be conducted to assess the development of endpoints over time. Multivariate Imputation will be used to handle missing data.

Also, outcome data will be collected for patients and relatives: the cost-effectiveness of care of dying patients, including the volunteer training programme, will be assessed by calculating the incremental cost-effectiveness ratio (ICER). The ICER is defined as the difference in total healthcare costs in the intervention group (including the costs of performing the intervention) compared with

the total costs in the control group, divided by the difference in outcomes.[36] Outcomes will also be measured as QALYs using the EQ-5D health-related quality of life questionnaire. We will also use the capabilities approach (ICECAP). The potential impact of the programme beyond the trial period will be extrapolated using modelling techniques.

### Qualitative data

Thematic analysis,[36] with an iterative approach to coding and analysis. For example, what was said by one participant may inform or shed light on the words of another participant. Analysis will focus on the 'substance of the interview' to interpret 'meanings and perceptions' created and shared during a conversation. Analysis of the data will explore the ideas, assumptions or concepts underpinning what participants are saying. Words or phrases will be coded at latent and semantic level to 'label' meaning identified by those words/phrases. 'Themes' will then be developed from these codes around a 'central observation' or concept to reflect patterns of shared meaning.[37] For example, to capture 'something important' about how these participants make sense of, and use, the ITVPal Programme within the context of end-of-life care.

Analysis will be conducted within a constructionist paradigm, which asserts that meaning and experience are socially produced. Stages of analysis will be iterative and cyclical rather than linear and analysis may move back and forth between stages. In summary, these principles are[36]:

► Familiarisation and immersion with the data: repeated reading of the interviews.
► Generation of initial codes: annotations of initial thoughts against items/categories of text.
► Creating themes from codes: the minor items/categories that have initially been identified from the text will then be interrogated to generate overall 'themes'. Engaging with relevant published literature (theoretical perspectives and relevant research) should further enhance the evolving interpretation.
► Reviewing themes: ensure all collated extracts of text form a coherent pattern, reorganising and refining themes as required.
► Defining and naming themes: to define the 'essence' of what a theme is about.
► Reporting the analysis: a narrative 'story' of the data to provide a concise, coherent, logical account.

### Cost–utility ratio

This is an exploratory objective (for which a specific design has not been applied). The cost–utility ratio will be estimated by dividing the total cost by the sum of the potential gains expressed in QALYs.

### ETHICS AND DISSEMINATION

The study will be carried out according to the Good Clinical Practice guidelines of the Declaration of Helsinki.[38]

Informed consent is obtained from each patient to take part in the study and for their clinical records to be reviewed. This project was approved by the corresponding ethics review board of the Malaga Provincial Research Ethics Committee on 25 February 2021.

The database is hosted on the secure REDCap platform at the University of Málaga, Spain. REDCap is an important tool for data access, linkages and mobilisation. On agreeing to participate in the project, clinics receive a random identification code, and the principal investigator and statistics team is blind to the coding.

All results and conclusions drawn from this project will be disseminated through local presentations, national or international meetings, academic publications and feedback to project participants.

In addition, as this project is funded by the La Caixa Foundation, the main results of the study will also be published for dissemination to the general public in different articles uploaded on its website.

### DISCUSSION

The COVID-19 pandemic is changing the way we are interacting, launching us seemingly overnight from the real world into the virtual world, and seeking ways to address the challenges of social distance.[39] Face-to-face support of PC volunteers has temporarily been suspended, so creativity and innovative solutions are needed to continue providing this care and support. The use of information and communication technologies and, specifically, NT, allows us to rethink the way volunteers act by integrating these resources to create 'tech-volunteers' that will attend to their needs in a different way and also support new needs that have emerged more clearly now due to social distance, with a broader reach.[40] The United Nations Online Volunteer Programme states: 'Online volunteering is fast, easy—and most of all, effective when skilled, passionate individuals join forces online with great organisations, everyone wins' but no robust study has been carried out to support this statement. In this way, this study will bring the real evidence toward NT integration as a useful tool not only to facilitate communication between volunteers and patients/relatives, but also to turn NT into instruments that support daily living and enhance care, when in the hands of trained tech-volunteers.

Although this new volunteer programme will be implemented during the pandemic to analyse the effects on patients and their relatives, we hope that training in NT will make it easier for PC volunteers to offer new skills and increase the satisfaction of the people they accompany.

The implementation of this study has some limitations that we will try to control for and take into account. For example, in pragmatic cluster randomised controlled trials, the first bias to take into account is the selection bias, as the randomisation is usually applied in the second level, before recruitment of the first-level participants, which is where the intervention impact is

measured. However, there are strategies that minimise this bias. First, the selection and follow-up of patients will be carried out by an external researcher who will have no knowledge of volunteer randomisation. Second, the effect of the intervention over recruitment can lead to different recruitment indexes, depending on the groups, due to the possibility that volunteers will be allocated a group in which they do not feel comfortable (eg, a volunteer who does not like the NT—technophobe—and will be allocated the intervention group) could be less motivated to participate. This issue will be taken into account in the analysis and it will be a result of the implementation process. There could be measurement bias due to mistakes in recording of some variables or interviewer bias due to the administration of the questionnaires. To minimise these biases, interviewers will have been trained previously to ensure that visits will be as homogeneous as possible. A data collection booklet and a manual that explains how each variable is measured will be designed.

Another bias that has already been mentioned is the involvement of volunteers in the year they will collaborate in ITV-Pal. For this reason, more volunteers will be trained. But also the number of patients/volunteers needed to achieve a representative sample and draw conclusions will be difficult to achieve in the 2 years of piloting the study.

Maybe the principal risk in this study will be that we will not be able to attend to the different needs of patients/relatives. In this case, we will try to adapt by offering other volunteers who may be able to give a better response to this need.

Due to the fact that in this study, we want to assess the implementation of an intervention programme, all these issues that we might find during the process could contribute to an improvement of the programme and would be useful for new research and the implementation of new interventions in the future.

**Author affiliations**
¹CUDECA Institute for Training and Research in Palliative Care, Málaga, Málaga, Spain
²Instituto de Investigación Biomédica de Málaga-IBIMA Group C08: Pharma economy: Clinical and economic evaluation of medication and Palliative Care, Málaga, Málaga, Spain
³Department of Social Psychology, Social Work, Social Anthropology and East Asian Studies, Faculty of Psychology and Speech Therapy, University of Malaga, Málaga, Málaga, Spain
⁴Instituto de Investigación Biomédica de Málaga-IBIMA Group CA15: Palliative Care, Málaga, Málaga, Spain

**Acknowledgements** We would first of all like to thank Fundación La Caixa for supporting this project and for enabling us to make it a reality. We would also like to thank all the patients, relatives, professionals and volunteers who have participated in the beginning of this project and have allowed this protocol to become a reality.

**Contributors** EV-M, PB-F and IR-T were in charge of project conception and protocol writing. PB-F focused on the methodological part and EV-M and IR-T on the other parts. MLM-R revised the protocol for its final version. EV-M and PB-F have produced the final version of this article and it has been reviewed by the rest of them. All authors have read and approved the final version of the manuscript. All authors (EV-M, PB-F, IR-T, MLM-R, RG-G and HC-B) have contributed to the development and review of the drafts that were used to submit this project and secure the grant.

**Funding** This work was supported by 'la Caixa' Foundation (SR20-00841).

**Competing interests** None declared.

**Patient and public involvement** Patients and/or the public were not involved in the design, or conduct, or reporting, or dissemination plans of this research.

**Patient consent for publication** Not applicable.

**Provenance and peer review** Not commissioned; externally peer reviewed.

**ORCID iDs**
Eva Víbora-Martín http://orcid.org/0000-0003-2680-8019
Rafael Gómez-García http://orcid.org/0000-0003-3904-8733

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
