## [Reviewer comments · BMJ Open]

ARTICLE DETAILS

TITLE (PROVISIONAL)	ITV-Pal Programme: Protocol of evaluation of the Implementation of Tech-Volunteer Programme in Palliative Care Services
AUTHORS	Barnestein-Fonseca, Pilar; VÍbora Martín, Eva; Ruiz-Torreras, Inmaculada; Chapinal-Bascón, Helena; Martín-Roselló, Maria; Gómez García, Rafael

VERSION 1 – REVIEW

REVIEWER	Radhika R Pai Manipal College of Nursing, Fundamentals of Nursing
REVIEW RETURNED	06-Sep-2022

GENERAL COMMENTS	The reviewer provided a marked copy with additional comments. Please contact the publisher for full details.
--

REVIEWER	Atsede Aregay Monash University, Nursing and Midwifery
REVIEW RETURNED	09-Sep-2022

GENERAL COMMENTS	Reviewer comments The study protocol is well described. The abstract, introduction, methods and discussion are well organized. However, some of the paragraphs and sentences have grammatical and punctuation errors. Most of the sentences were not supported by the references. Abstract Strength and limitation Important points are mentioned in the strength and limitation section. However, the second and third strength are already mentioned in the first sentence. Introduction In the second sentences (line 52) of the first paragraph should be supported with the reference. In the next sentence, you mentioned that '... expected to die with one of these causes.' My question here is, what are the causes of death? You did not mention them in the previous sentence. In the second paragraph of the second sentence 'pharmacological ... a patient', the sentence lacks clarity. In this paragraph, sentences 4 and 5 are not supported with the references. In the last sentence of this paragraph, there is a punctuation and grammatical error. For this reason, the sentence lacks clarity. Similarly, the third paragraph of the first sentence has grammatical error. For example, 'is of' In the last sentence of the paragraph, there is a need of space between 'can' and 'be' rather than 'canbe'. Why is the next sentence starting from 'in addition ...' be a separate paragraph? I suggest being merged with the previous paragraph because the idea is continuing from the previous sentence. In addition, the sentence needs to have a full stop at the end. In the next paragraph, please have a look the two words
---

'suchresources'. One thing there is a need of space between the words and the second thing is the word 'such' is not properly written.

In the next paragraph that starts in line 55, why some of the words are bold? The last sentence is too long which is difficult to follow it. In the paragraph that starts with 'Although volunteers contribute millions of hours ...' what is millions of hours of work mean? In addition, the next sentence starts with 'As more research ...' is too long and difficult to understand the meaning of the sentence. I would suggest revising this paragraph to make easy and clear for the reader to understand the flow of sentence.

What is 'the global aim of the study' mean? In the specific objectives, a space is required in 'patients,relatives'

Methodology

Please have a look the word 'combination' in the first sentence.

Why do you underline some of the words in this section?

In A quantitative approach, at the end of the sentence, a space required in 'volunteersof'

Under the before-after design, the words need a space 'thosemeasured'

Under the quantitative approach. The sentence that described about the Pragmatic cluster randomized clinical trial should be supported by the reference.

Setting

Please add 's' because it is more than one setting. It will be relevant if you clarify more about the settings.

Exclusion criteria

'Subjects who have no specific conditions reducing their physical ability to use the devices, for example, visual, hearing, or motor impairments will be excluded from the study.' Does this sentence correct? Are you excluding those who have no visual, hearing or motor impairment? Does the subject represent from the parent/relative or all including HCP?

You already mentioned at the previous sentence patients/families included in the study. Why is patient not involved? What is that mean? What type of patient is not involved?

Intervention

The second sentence that started with 'the topic that will cover ...' is too long. Could you please split into two sentences?

Variables

Patients/relatives/volunteers/HCP: the points mentioned are not clear. Could you please describe them?

The process evaluation

Some of the sentences started with 'whether the intervention was delivered ...' are not complete. And why are the first letter are capitalised after the (:)? The whole paragraph is difficult to understand because of incomplete sentences.

Could you please use a consistent term either patients/relatives or patient/families throughout the paper?

Volunteers

A space is required in the 'experienceof'. Please add 's' for patient/relative because in most of the paper I have seen without s.

Data analysis

A space required in 'ifp < 0.05.'

The sentence that defines 'The ICER is defined as ...' should supported by the reference. If that is a definition, you also need to quote it.

	The type and steps of analysis should be supported by the reference. Ethical and Dissemination. Grammatical error in the sentence 'informed consent is obtained' Discussion Under the sentence '...effective. When ...' if the quotation is in one, no need (.) between the sentences. What is CRF? Please try to use full name before you use an abbreviation. The paragraph before the acknowledgement has a grammatical and punctuation errors 'Due to in this study...' None of the sentences or paragraphs was supported by any references in the discussion. Could you please clarify why you are not cited the references? References Why is number 9 in the list of references. Is it required to use [1], [2], [3] ...? What are the supplementary materials? Are the list of references and figure 1? If that is so why you start 9 to the reference and 10 to the figure 1? Figure 1. it looks good. But I think you need to upload as a picture so that the arrow and others will not be misplaced. General comments The date of the study was not mentioned in the manuscript. When will the study will be conducted? How do you select 140 patients/relatives and 20 volunteers? Could please clarify that?
--	---

REVIEWER	Terri Kean University of Prince Edward Island, Faculty of Nursing (retired)
REVIEW RETURNED	28-Nov-2022

GENERAL COMMENTS	ARTICLE: ITV-Pal Programme: Protocol of evaluation of the Implementation of Tech-Volunteer Programme in Palliative Care Services REVIEWER: Terri Kean Thank you for inviting me to review this protocol article. Increased need for home and hospital based palliative care services couples with limited resources (e.g., human, financial, and time) necessitates the use of innovative approaches to support individuals and those who are significant to them at the end of life. This research is timely and much needed. The comments below seek only to gain further understanding and clarify the protocol of this important work. Note: Supporting documents (e.g., Consent form, Questions to determine technological profile of participants, Interview Guide, Tools listed under Variables section) did not accompany the protocol and therefore could not be reviewed In General
--

	 [ ] The article represents important work [ ] Added details and revised flow in some areas would enhance readability [ ] The lack of detail would make it difficult to duplicate this study as written [ ] Tests for quantitative testing were included but not described [ ] References (32 total) - 18 (56.25%) dated 2012 or earlier; 9/32 (28.12%) dated between 2013- 2017; and 5/32 (15.62%) dated 2018 to present (perhaps there is current literature on the use of NT by different groups in other areas of health) ABSTRACT:  [ ] It may be helpful to review the abstract introduction to ensure sentence flow and word clarity  o For example, the second sentence in paragraph 1 seems to be missing something to link it to the first. Perhaps something like ... Volunteer support represents an important and necessary community resource for individuals experiencing life-threatening illness and their caregivers. o It may be just different terminology but line 49 p. 2 – “review of bibliography” could be expressed as review of the literature [ ] Incongruencies seem to arise between the abstract and the protocol itself. For example line 59 (p.2) of the abstract indicates the researchers will conduct “a pilot study of 20 volunteers and 70 patients over two years”. However, the text of the protocol (line 48 p.6 within the Participant section) identifies “140 patients/relatives and 20 volunteers were required”. [ ] Within the abstract (lines 53-58 p.2) the researchers state “we intend to study the state of art of technophilia and technophobia, and according to the results, implement and evaluate a volunteer training programme in the use of NT to support patients dying at home and their relatives”. This is repeated in lines 12-15 of the introduction (p.5) yet the global aim or specific objectives do not reflect a connection to technophilia and technophobia. The study is designed to explore the “need and usefulness of NT...” (specific object # 1); however, it is uncertain if this objective capture the participants’ beliefs and/or attitudes toward NT that would lead to an understanding of the technophilia and technophobia among the research population. INTRODUCTION:  [ ] Objectives # 1 and # 2 (line 22-27 p. 5) support the design/development of a volunteer training program yet line 18-22 (p. 5) indicates the global aim of this study is to implement and evaluate a volunteer training programme.  o Should the global aim be expanded to include the design, implementation, and evaluation of a volunteer training programme? METHODOLOGY Study Design Details on how the following will be achieved were not available in the protocol  [ ] Objective 1 - Explore the need and usefulness of NT from the point of view of patients, relatives, volunteers and health care professionals (HCP) in PC and describe their technological profile. [ ] Objective 2 - Design a tech-volunteer curriculum and implement a tech- volunteer training programme. Setting  [ ] To clarify – Will some patients/caregivers be participating from home and others be participating from the hospital?
--	--

	Participants  [ ] Can you clarify this sentence “Subjects who have no specific conditions reducing their physical ability to use the devices, for example, visual, hearing, or motor impairments will be excluded from the study.”? [ ] It seems to suggest participants must have a specific condition that reduces their physical ability to use the devices e.g., hearing, vision, or physical impairment [ ] If so, how are the researchers determining physical suitability – Medical records? Physical exam? Participant self-reporting? [ ] Is there any criteria that would exclude a patient from participating other than death? For example, what if a patient’s condition deteriorates to such a point that they can not participate in the study but are still alive? [ ] How is competency to consent and/or participate being determined among a population who may be receiving significant analgesia as part of their palliative care plan? [ ] Inclusion criteria for HCPs – Who falls under the term HCP? Are these physicians (general practitioners or specialists), nurses, dietitians, pharmacists, respiratory therapists, physical therapists, and/or occupational therapists? [ ] Inclusion criteria for volunteers – Can you clarify what “The inclusion criteria for volunteers will be an interview with the Volunteer Department...” means [ ] Does this mean the volunteers will be recruited via an interview with the Volunteer department or only volunteers from a specific volunteer centre will be recruited? [ ] What will happen if a volunteer must unexpectedly leave the study or there is a change of volunteer for a patient during the study? Recruitment – no details on how possible participants will be recruited Patient and Public Involvement  [ ] Line 10-12 (p.7) indicates “no patient involvement”; however, it appears patients are being recruited as possible participants [ ] Public involvement not commented on – would the volunteers be considered “public involvement”? Intervention First phase – How will volunteers be trained? individual, group, `classroom, self-study, face-to face, online? Ten training sessions are planned – what is the timeframe for each training session? How long is each training session? How long will it take to complete 10 sessions? Control Group – What is the standard training volunteers currently receive? Variables (Patients, Relatives, Volunteers, and HCPs)  [ ] In this section a number of tools/scales are listed [ ] The purpose of the EuroQol-5D-5L is described under the Patient variables section [ ] A discussion of the variables being measured is not included in the article but would be helpful to identify the variables each measures and the reliability (validity/specificity of each tool where available) Data Collection and Follow-up  [ ] Researchers indicate focus group or in-depth interviews (line 40-42 p. 8) but details of focus group are not included nor mentioned in this section Patient/Families  [ ] Will be invited to fill in at least 3 short questionnaires –
--	---

	which questionnaires? How will this be decided? Are the questionnaires the tools listed under variables? Is there a chance the participants may be asked to fill out more than three short questionnaires (the protocol say at least three – does this suggest more than three is possible?)  [ ] Perhaps clarify – participants will be asked to fill out questionnaires every month as long as the patient remains under PC care or until death - ? for the duration of the study – 2 years [ ] Maximum of 5 patients/families will be asked to participate in in-depth interviews at various points throughout the study  [ ] Is there a minimum number of patients/families that will be included in in-depth interviews (max of 5 indicated)? [ ] Duration of in-depth interview? [ ] Is there a specific timeframe for when patients/families will be asked to participate in in- depth interviews? [ ] How will the interviews be recorded? Recorded? [ ] Process for transcribing/reviewing interviews? Volunteers  [ ] Which questionnaires with the volunteers fill out? Are these the tools listed under variables? [ ] Maximum of 5 volunteers will be asked to participate in in-depth interviews at various points throughout the study  [ ] Is there a minimum number of volunteers that will be included in in-depth interviews (max of 5 indicated)? [ ] Duration of in-depth interview? [ ] Is there a specific timeframe for when volunteers will be asked to participate in in-depth interviews? [ ] How will the interviews be recorded? Recorded? [ ] Process for transcribing/reviewing interviews? HCPs  [ ] Maximum of 5 HCPs will be asked to participate in in-depth interviews at various points throughout the study  [ ] Is there a minimum number of HCPs that will be included in in-depth interviews (max of 5 indicated)? [ ] Duration of in-depth interview? [ ] Is there a specific timeframe for when volunteers will be asked to participate in in-depth interviews? [ ] How will the interviews be recorded? Recorded? [ ] Process for transcribing/reviewing interviews? Data Analysis  [ ] It would be helpful to note whether thematic analysis will be completed manually or with the use of software designed for this purpose (e.g., NVivo)
--	---

VERSION 1 – AUTHOR RESPONSE

Reviewer comments	Author' s response
Some of the paragraphs and sentences have	The manuscript has been review by a English native to improve the expressions, grammatical and punctuation errors.
Strength and limitation: Important points are mentioned in the strength and limitation section. However, the second andthird strength are already mentioned in the first sentence.	Second and third strengths have been added to the first point.

In the second sentences (line 52) of the first paragraph should be supported with the reference.	Reference added. Line: 32
In the next sentence, you mentioned that ‘... expected to die with one of these causes.’ My question here is, what are the causes of death? You did not mention them in the previous sentence.	Causes added. Lines: 34-35
In the second paragraph of the second sentence ‘pharmacological ... a patient’, the sentence lacks clarity.	The sentence has been modified to clarify. Line: 38
In this paragraph, sentences 4 and 5 are not supported with the references. In the last sentence of this paragraph, there is a punctuation and grammatical error. For this reason, the sentence lacks clarity. Similarly, the third paragraph of the first sentence has grammatical error. For example, ‘is of’	Reference added. Line: 3
In the last sentence of this paragraph, there is a punctuation and grammatical error. For this reason, the sentence lacks clarity.	Reviewed
Similarly, the third paragraph of the first sentence has grammatical error. For example, ‘is of’	Reviewed
In the last sentence of the paragraph, there is a need of space between ‘can’ and ‘be’ rather than ‘canbe’.	DONE
Why is the next sentence starting from ‘in addition ...’ be a separate paragraph? I suggest being merged with the previous paragraph because the idea is continuing from the previous sentence. In addition, the sentence needs to have a full stop at the end.	DONE
In the next paragraph, please have a look the two words ‘suchresources’. One thing there is a need of space between the words and the second thing is the word ‘such’ is not properly written.	DONE
In the next paragraph that starts in line 55, why some of the words are bold?	Some words are in bold to highlight the different tasks that the use of NTs can

The last sentence is toolong which is difficult to follow it.	facilitate in everyday life. This is for clarification, if you think it may lead to confusion. It can be modified. Sentence modified. Lines: 34-36
In the paragraph that starts with 'Although volunteers contribute millions of hours ...' what is millions of hours of work mean?	Sentence modified. Lines: 1-2
In addition, the next sentence starts with 'As more research ...' istoo long and difficult to understand the meaning of the sentence. I would suggest revising this paragraph to make easy and clear for the reader to understand the flow of sentence.	Sentence modified. Lines: 3-7
What is 'the global aim of the study' mean?	Rewritten. Lines: 9-15
In the specific objectives, a space is required in 'patients,relatives'	DONE
Please have a look the word 'combination' in the first sentence.	DONE
Why do you underline some of the words in this section?	To clarify different subheadings. This is for clarification, if you think it may lead to confusion. It can be modified.
In A quantitative approach, at the end of the sentence, a space required in 'volunteersof'	DONE
Under the before-after design, the words need a space 'thosemeasured'	DONE
Under the quantitative approach. The sentence that described about the Pragmatic cluster randomized clinical trial should be supported by the reference.	References added. Line: 33

Please add 's' because it is more than one setting.	DONE
It will be relevant if you clarify more about the settings.	Sentence modified. Lines: 19-27
'Subjects who have no specific conditions reducing their physical ability to use the devices, for example, visual, hearing, or motor impairments will be excluded from the study.' Does this sentence correct? Are you excluding those who have no visual, hearing or motor impairment?	Erratum. It has already been amended to read the opposite.
Does the subject represent from the parent/relative or all including HCP? You already mentioned at the previous sentence patients/families included in the study. Why is patient not involved? What is that mean? What type of patient is not involved?	We are sorry but we do not understand the question. The subjects are patients and/or relatives, as we mentioned in the Participant section. HCP are included to measure the implementation of the programme. The patients and/or relatives involved are those who meet the inclusion criteria.
The second sentence that started with 'the topic that will cover ...' is too long. Could you please split into two sentences?	Sentence modified. Lines: 17-20
Patients/relatives/volunteers/HCP: the points mentioned are not clear. Could you please describe them?	Added new sentence. Lines: 5-6 Describe each questionnaire.
Some of the sentences started with 'whether the intervention was delivered ...' are not complete. And why are the first letter are capitalised after the (:)? The whole paragraph is difficult to understand because of incomplete sentences.	Rewritten. Lines: 6-12
Could you please use a consistent term either patients/relatives or patient/families throughout the paper?	DONE

A space is required in the 'experienceof'. Please add 's' for patient/relative because in most of the paper I have seen without s.	DONE
A space required in 'ifp < 0.05.'	Modified. Line: 13
The sentence that defines 'The ICER is defined as ...' should supported by the reference. If that is a definition, you also need to quote it.	Cited Line: 20
The type and steps of analysis should be supported by the reference.	Cited Line: 2
Under the sentence '...effective. When ...' if the quotation is in one, no need (.) between the sentences.	DONE
What is CRF? Please try to use full name before you use an abbreviation.	DONE
None of the sentences or paragraphs was supported by any references in the discussion. Could you please clarify why you are not cited the references?	Added. Lines: 35, 4
Why is number 9 in the list of references. Is it required to use [1], [2], [3] ...? What are the supplementary materials? Are the list of references and figure 1? If that is so why you start 9 to the reference and 10 to the figure 1?	The previous list has been removed in case it was confusing for the reader.
Figure 1. it looks good. But I think you need to upload as a picture so that the arrow and others will not be misplaced.	It will be re-uploaded as a jpg image
The date of the study was not mentioned in the	Rewritten.

manuscript. When will the study will be conducted?	Lines: 15-18
How do you select 140 patients/relatives and 20 volunteers? Could please clarify that?	Explained. Lines: 30-1
Reviewer comments	Author' s response
please specify the causes	Causes added. Lines: 34-35
please go through the methodology of mixed method research. You may use a relevant mixed method with the schematic representation of the design.	We do not understand what means
these can be included in the data collection	We consider that this part is convenient in methodology part because we specify the details of cluster design in our study
can explain these under data collection plan	We consider that this part is convenient in methodology part because we specify how we are going to develop the qualitative design in our study
write intervention detail here for both intervention and control group. include random sampling details, concealment, random assignment, blinding details	We believe that the order we have established is correct but if you prefer we can modify it by moving the data collection and follow up section after study design.
please specify from where these differences were taken or cite the study which was used for the sample size calculation	Reference added. Line: 33
write these session detail including the time	We have indicated the time but the details of the sessions will be indicated in later papers because they have been modified according to the needs of the volunteers.

tell here what exactly happens to patients and relatives or what your are expecting their involvement	Explained. Lines: 23-27
write in detail regarding each tool used in the study, including the interpretation of the scores, reliability and validity details	Added. Further in-depth details are explained in the bibliographical references.
these doesnt look like variables, they are tools or questionnaires used in the proposed study	In addition to the questionnaires, the socio-demographic variables of the study have been added. Lines: 30-4

VERSION 2 – REVIEW

REVIEWER	Radhika R Pai Manipal College of Nursing, Fundamentals of Nursing
REVIEW RETURNED	15-Feb-2023

GENERAL COMMENTS	The reviewer provided a marked copy with additional comments. Please contact the publisher for full details.
--

REVIEWER	Atsede Aregay Monash University, Nursing and Midwifery
REVIEW RETURNED	16-Feb-2023

GENERAL COMMENTS	Abstract: Why is the new technologies start as a new paragraph. I would recommend being continues after the volunteer support sentence. The sentence “volunteer support ...” is not completed. In what way is important community resource? As unfamiliar reader, It is relevant to describe the terms ‘technophilia’ and ‘technophobia’ In general, the introduction has a separated paragraph. I would suggest in summarising the relevant points in one paragraph. Methods and analysis: change the term ‘mix-methods’ □mixed-method. Please paraphrase the sentence. For example: “A mixed-method study design will be conducted. We will recruit 20 volunteers and 70 patients in two years”. Or “A mixed-method study design will be conducted among 20 volunteers and 70 patients in two years”. Not recommended to use abbreviation in the abstract. Can use please use the NT, PC, ... in the introduction and the rest of the manuscript. Ethics and dissemination The final sentence should be revised “... feedback to the participants and to clinics that will be involved in the study. ITV-Pa: you have used this for the first time. It will be important if you describe first then you can use that later. Key words: I would suggest including mixed-method study as a key word if no limit of words. Strength and limitation How are you planning to solve the first limitation? If volunteers are not comfortable in the assigned place. I have not seen an intervention related in the methods section in
--

	the abstract. Why? Introduction Is not it cultural too? Care for the dying is cultural or context based, I think. Line 3: Reference is required to the sentence 'Dying is a social process' similarly. In line 7 volunteer support... (refe). Reference is required for line 18: NT... Again you have used these terms 'technophilia and technophobia' without describing what they are to the reader. Could you please paraphrase the beginning of the last paragraph as "The aim of this study will be to evaluate the Could you please write the full name of the abbreviations before you use them such as IT, HPC. And If the following sentence describe 'CUDECA Foundation' "In addition, the foundation also provides home care to more than 1,500 patients a year through 7 care 26 teams made up of a doctor, nurse, psychologist and social worker. In these cases, the volunteer 27 accompanies patients and relatives who wish them to do so." Please link the sentence into the top or just try to rewrite as "CUDECA Foundation provides home care ...". Patient and Public Involvement Still, this is not clear. Who is not involved? 'no patient involved' Qualitative data: Please try to include references for thematic analysis. You have mentioned the reference in another paragraph. It will be relevant to include in the first sentences too. The contents of figure 1 particularly those words in between the arrow are difficult to read. They are not as such visible. Why is a 'discussion' since the contents are limitation of study. I would suggest changing the heading of discussion into limitation of the study.
--	---

VERSION 2 – AUTHOR RESPONSE

Reviewer comments	Author' s response
Some of the paragraphs and sentences have	The manuscript has been review by a English native to improve the expressions, grammatical and punctuation errors.
Strength and limitation: Important points are mentioned in the strength and limitation section. However, the second andthird strength are already mentioned in the first sentence.	Second and third strengths have been added to the first point.
In the second sentences (line 52) of the first paragraph should be supported with the reference.	Reference added. Line: 32
In the next sentence, you mentioned that '... expected to die with one of these causes.' My question here is, what are the causes of death? You did not mention them in the previous sentence.	Causes added. Lines: 34-35

In the second paragraph of the second sentence 'pharmacological ... a patient', the sentence lacks clarity.	The sentence has been modified to clarify. Line: 38
In this paragraph, sentences 4 and 5 are not supported with the references. In the last sentence of this paragraph, there is a punctuation and grammatical error. For this reason, the sentence lacks clarity. Similarly, the third paragraph of the first sentence has grammatical error. For example, 'is of'	Reference added. Line: 3
In the last sentence of this paragraph, there is a punctuation and grammatical error. For this reason, the sentence ... lacks clarity.	Reviewed
Similarly, the third paragraph of the first sentence has grammatical error. For example, 'is of'	Reviewed
In the last sentence of the paragraph, there is a need of space between 'can' and 'be' rather than 'canbe'.	DONE
Why is the next sentence starting from 'in addition ...' be a separate paragraph? I suggest being merged with the previous paragraph because the idea is continuing from the previous sentence. In addition, the sentence needs to have a full stop at the end.	DONE
In the next paragraph, please have a look the two words 'suchresources'. One thing there is a needof space between the words and the second thing is the word 'such' is not properly written.	DONE
In the next paragraph that starts in line 55, why some of the words are bold? The last sentence is toolong which is difficult to follow it.	Some words are in bold to highlight the different tasks that the use of NTs can facilitate in everyday life. This is for clarification, if you think it may lead to confusion. It can be modified. Sentence modified. Lines: 34-36

In the paragraph that starts with 'Although volunteers contribute millions of hours ...' what is millions of hours of work mean?	Sentence modified. Lines: 1-2
In addition, the next sentence starts with 'As more research ...' isto long and difficult to understand the meaning of the sentence. I would suggest revising this paragraph to make easy and clear for the reader to understand the flow of sentence.	Sentence modified. Lines: 3-7
What is 'the global aim of the study' mean?	Rewritten. Lines: 9-15
In the specific objectives, a space is required in 'patients,relatives'	DONE
Please have a look the word 'combination' in the first sentence.	DONE
Why do you underline some of the words in this section?	To clarify different subheadings. This is for clarification, if you think it may lead to confusion. It can be modified.
In A quantitative approach, at the end of the sentence, a space required in 'volunteersof'	DONE
Under the before-after design, the words need a space 'thosemeasured'	DONE
Under the quantitative approach. The sentence that described about the Pragmatic cluster randomized clinical trial should be supported by the reference.	References added. Line: 33
Please add 's' because it is more than one setting.	DONE
It will be relevant if you clarify more about the settings.	Sentence modified. Lines: 19-27
'Subjects who have no specific conditions reducing their physical ability to use the devices,	

for example, visual, hearing, or motor impairments will be excluded from the study.’ Does this sentence correct? Are you excluding those who have no visual, hearing or motor impairment?	Erratum. It has already been amended to read the opposite.
Does the subject represent from the parent/relative or all including HCP? You already mentioned at the previous sentence patients/families included in the study. Why is patient not involved? What is that mean? What type of patient is not involved?	We are sorry but we do not understand the question. The subjects are patients and/or relatives, as we mentioned in the Participant section. HCP are included to measure the implementation of the programme. The patients and/or relatives involved are those who meet the inclusion criteria.
The second sentence that started with ‘the topic that will cover ...’ is too long. Could you please split into two sentences?	Sentence modified. Lines: 17-20
Patients/relatives/volunteers/HCP: the points mentioned are not clear. Could you please describe them?	Added new sentence. Lines: 5-6 Describe each questionnaire.
Some of the sentences started with ‘whether the intervention was delivered ...’ are not complete. And why are the first letter are capitalised after the (:)? The whole paragraph is difficult to understand because of incomplete sentences.	Rewritten. Lines: 6-12
Could you please use a consistent term either patients/relatives or patient/families throughout the paper?	DONE
A space is required in the ‘experienceof’. Please add ‘s’ for patient/relative because in most of the paper I have seen without s.	DONE
A space required in ‘ifp < 0.05.’	Modified. Line: 13
The sentence that defines ‘The ICER is defined as ...’ should supported by the reference. If that is a definition, you also need to quote it.	Cited Line: 20

The type and steps of analysis should be supported by the reference.	Cited Line: 2
Under the sentence ' ...effective. When ... ' if the quotation is in one, no need (.) between the sentences.	DONE
What is CRF? Please try to use full name before you use an abbreviation.	DONE
None of the sentences or paragraphs was supported by any references in the discussion. Could you please clarify why you are not cited the references?	Added. Lines: 35, 4
Why is number 9 in the list of references. Is it required to use [1], [2], [3] ...? What are the supplementary materials? Are the list of references and figure 1? If that is so why you start 9 to the reference and 10 to the figure 1?	The previous list has been removed in case it was confusing for the reader.
Figure 1. it looks good. But I think you need to upload as a picture so that the arrow and others will not be misplaced.	It will be re-uploaded as a jpg image
The date of the study was not mentioned in the manuscript. When will the study will be conducted?	Rewritten. Lines: 15-18
How do you select 140 patients/relatives and 20 volunteers? Could please clarify that?	Explained. Lines: 30-1
Reviewer comments	Author' s response
please specify the causes	Causes added.

	Lines: 34-35
please go through the methodology of mixed method research. You may use a relevant mixed method with the schematic representation of the design.	We do not understand what means
these can be included in the data collection	We consider that this part is convenient in methodology part because we specify the details of cluster design in our study
can explain these under data collection plan	We consider that this part is convenient in methodology part because we specify how we are going to develop the qualitative design in our study
write intervention detail here for both intervention and control group. include random sampling details, concealment, random assignment, blinding details	We believe that the order we have established is correct but if you prefer we can modify it by moving the data collection and follow up section after study design.
please specify from where these differences were taken or cite the study which was used for the sample size calculation	Reference added. Line: 33
write these session detail including the time	We have indicated the time but the details of the sessions will be indicated in later papers because they have been modified according to the needs of the volunteers.
tell here what exactly happens to patients and relatives or what your are expecting their involvement	Explained. Lines: 23-27
write in detail regarding each tool used in the study, including the interpretation of the scores, reliability and validity details	Added. Further in-depth details are explained in the bibliographical references.
these doesnt look like variables, they are tools or questionnaires used in the proposed study	In addition to the questionnaires, the socio-demographic variables of the study have been added.

	Lines: 30-4
--	-------------

VERSION 3 – REVIEW

REVIEWER	Atsede Aregay Monash University, Nursing and Midwifery
REVIEW RETURNED	30-Mar-2023

GENERAL COMMENTS	Introduction: the first and second sentences are not linked. For example, Volunteers are important resource for what....? I will suggest starting from the second sentence. However, still the second sentence is not completed. The sentence from line 36 to 39 should continue from the above sentence. However, I would suggest modifying the sentence. For example ‘... when associating palliative care volunteering and New technologies. This study aims to evaluate ...’ The introduction part is too much. In the abstract of the introduction, the author should focus on the main objective and what is missing and what this protocol planning to contribute. Not too detail description. Is the discussion required in this stage? Could you please clarify why it is included? There are a number of grammatical and punctuation errors in the sentences and paragraphs. I would suggest revising the paragraphs and each sentence of the paragraph. In addition, still there are a number of sentences that are not supported by the references or not properly cited. Most of the references are old. Could you please consider using updated articles or books.
---

VERSION 3 – AUTHOR RESPONSE

Reviewer’s comments	Author’ s response
Introduction: the first and second sentences are not linked. For example, Volunteers are important resource for what....? I will suggest starting from the second sentence. However, still the second sentence is not completed. The sentence from line 36 to 39 should continue from the above sentence. However, I would suggest modifying the sentence. For example ‘... when associating palliative care volunteering and New technologies. This study aims to evaluate ...’ The introduction part is too much. In the abstract of the introduction, the author should focus on the main objective and what is missing and what this protocol planning to contribute. Not too detail description.	Changed. Page: 1. Lines: 29-35

Is the discussion required in this stage? Could you please clarify why it is included?	Yes, we believe that the discussion section is very important, even if it is a protocol, as this section talks about the potential impact of the research to be done and the applicability of the study with its strengths and limitations. This is a requirement for all protocols submitted for funding and is a very important section for the possibility of funding. Therefore, we believe that this part is very important to fully understand the contribution and feasibility of the study.
There are a number of grammatical and punctuation errors in the sentences and paragraphs. I would suggest revising the paragraphs and each sentence of the paragraph.	The manuscript has been rechecked by a native English reviewer to look for the errors you refer to. We hope to have located the errors you refer to, although we have not found too many during editing. If you notice that any of them have not been taken into account, please tell us exactly what you are referring to, it is possible that it is just a difference in the way of expression and that is why our English reviewer has not found it.

In addition, still there are a number of sentences that are not supported by the references or not properly cited.	We have rechecked the bibliographic citations in the text and we are at the same point as before. We consider that all sentences and paragraphs are supported by bibliographies, and if any are not supported before the dot separating the sentences, maybe it is because the citation is at the end of the sentences or of the paragraph containing these sentences and which is complete corresponding to this citation. I don't know if this is what you are referring to specifically. In any case, if you think it is really important, please tell us exactly where they are.
Most of the references are old. Could you please consider using updated articles or books.	We have conducted a new search, because we are preparing another paper, and we have not obtained more updated bibliography. In any case, if you have any bibliography more updated, please send us to update the paper.